# Development of 3D-Printed Orthopedic Insoles for Patients with Diabetes and Evaluation with Electronic Pressure Sensors

**Juan Zuñiga [1], Miguel Moscoso [2], Pierre G. Padilla-Huamantinco [3,4], Maria Lazo-Porras [5], Janeth Tenorio-Mucha [5], Wendy Padilla-Huamantinco [6] and Jean Pierre Tincopa [7,*]**

1   Departamento de Ingeniería, Facultad de Ciencias y Filosofía, Universidad Peruana Cayetano Heredia, Lima 15102, Peru
2   Escuela de Medicina Humana, Universidad Continental, Lima 15046, Peru
3   Health Innovation Lab, Institute of Tropical Medicine Alexander von Humboldt, Universidad Peruana Cayetano Heredia, Lima 15102, Peru
4   Institute for Biological and Medical Engineering, Schools of Engineering, Medicine, and Biological Sciences, Pontificia Universidad Católica de Chile, Santiago 8331150, Chile
5   CONEVID, Unidad de Conocimiento y Evidencia, Facultad de Medicina "Alberto Hurtado", Universidad Peruana Cayetano Heredia, Lima 15102, Peru
6   Facultad de Ciencias de la Salud, Universidad Peruana de Ciencias Aplicadas, Lima 15023, Peru
7   Digital Transformation Research Center, Norbert Wiener University, Lima 15046, Peru
*   Correspondence: jeanpierre.tincopa@uwiener.edu.pe

**Abstract:** The correct distribution of loads on foot, known as plantar pressures, is a relevant parameter for evaluating the evolution of some diseases. Anomalies can lead to pain and discomfort in other body parts. Diabetes changes foot tissues and compromises biomechanics, resulting in ulcers and, eventually, amputation. Customized insoles allow the redistribution of plantar pressures and are a complementary strategy to diabetes management. Nowadays, scanning and 3D printing technology can generate faster and more accurate customized insoles opening new opportunities for local medical device development. This study reports the development of 3D-printed insoles using two polymers, thermoplastic polyether-polyurethane and thermoplastic polyurethane polyester-based polymer, and the evaluation of plantar pressure distribution in walk trials using a clinical protocol and low-cost electronic system. The two 3D-printed insoles performed as well as a standard insole. No significant difference was found in average peak pressure distribution. The digital manufacturing workflow of customized insoles can be implemented in middle-income countries. Three-dimensionally printed insoles have the potential for diabetes management, and further material evaluations are needed before using them in health facilities.

**Keywords:** insoles; 3D printing; plantar pressure; electronic sensor

## 1. Introduction

In the last 20 years, the number of people with diabetes has doubled [1]. Currently, some 400 million people worldwide suffer from the disease; it is estimated that these numbers will reach 600 million in the next 15 to 20 years [2]. Diabetes mellitus has complications that increase disability and mortality, and the impact in low- and middle-income countries is even higher compared to high-income countries [3]. It is estimated that 50% of people with diabetes will suffer from diabetic neuropathy throughout their lives [4]. Eventually, neuropathy can lead to further complications resulting in amputation or even mortality. The plantar pressure of the foot is the pressure field that acts between the foot and the support surface during daily locomotor activities. Measuring this parameter is of the utmost importance; it helps in the design of orthopedic shoes or insoles to improve plantar pressure and prevent the generation of injuries [5]. In the case of people with diabetes, plantar pressures are used to assess the risk of ulceration [6]. In healthy people, the soft tissue of the sole has a specialized structure whose function is to dissipate stress

and absorb the impact of a foot strike [7]. However, in people with diabetes, their soles are compromised by changes in foot tissue affecting the functionality and mechanical integrity [8]. As a consequence, there is an increase in plantar pressures in an uneven way. This increase impedes oxygen supply and increases the risk of ulceration [9]. Similarly, biomechanical misalignments in the foot region can lead to abnormal loads in different parts of the foot, thus varying the plantar pressure, which can consequently generate heel, knee, and back pain. The standard treatment method is prefabricated insoles, but custom insoles can also relieve this plantar pressure [10].

In the last decade, 3D printing has become popular worldwide. Among different additive manufacturing technologies, fused deposition modeling (FDM) is one of the most widely used and can be applied for insole fabrication. Some studies have demonstrated the designing and manufacturing of insoles using 3D scanning, digital modeling, and 3D printing technology [11]. This approach is helpful because highly customized insoles can be made in a shorter time than traditional methods. The plantar pressure distribution and the contact area are different in each patient; therefore, the plantar pressure tends to have many variations. Three-dimensionally printed insoles can meet these individual needs for clinical treatment with different materials, shapes, and internal patterns [12]. When new insoles are prescribed, the plantar pressure parameter allows specialists to verify appropriate device fabrication and manage the treatment. In-shoe methods using electronic systems were developed to measure this parameter, and the first system was reported in the 1960s [13]. Over the years, several in-shoe methods have been implemented and are commonly used for research and clinical practice [14]. These electronic systems are mainly based on flexible sensors that measure pressure between the foot and the shoe. Currently, various sensors such as polymer and piezoelectric sensors can be used. The Force-Sensitive Resistor (FSR) is a polymer-based device whose internal resistance varies when different magnitudes of force are exerted on its surface. These sensors have flexible parts with printed semiconductors in conjunction with interdigital electrodes that consist of two individually addressable microelectrode array strips [15] in their composition, so they are usually easier to attach to different surfaces. Piezoelectric sensors can generate an electrical charge under mechanical stress. Variations in its structure produce tiny electric currents that can be transformed into voltage spikes through resistance or inducing electric potential that can change its dimensions [16].

Previous studies have reported using and evaluating 3D-printed insoles. In Korea, 15 people were evaluated using 3D-printed insoles in active walks. It was found that insoles changed the trajectory of the center of pressure (COP) during walking fast, thus improving the load distribution on foot during the stance phase [17]. Another study focused on analyzing the changes in plantar pressure during a light walk. The feet of the selected subjects were scanned, and customized insoles were designed. These insoles in contact with other surfaces were tested in three different scenarios: an original shoe insole, a customized insole of ethylene-vinyl acetate, and a 3D-printed anatomical insole. The results demonstrated that customized insoles influence plantar pressure distribution, mainly in people with flat feet or high-arched foot deformities. [11].

From a mechanical perspective, some authors have evaluated the design and materials of customized insoles using finite element analysis, defining the static and dynamic loads that can act on the insole [18]. There is a report where the infill (filling pattern) of 3D-printed insoles with semi-rigid materials was analyzed using different tests and measuring parameters such as traction, bending, and hardness resistance [19]. In China, some researchers explored the effect of using customized 3D-printed insoles in patients with symptomatic flat feet compared to traditional prefabricated insoles. The findings demonstrated that 3D-printed insoles reduced pressure on the metatarsals by distributing them over the midfoot area [10]. Other strategies of measurement have been shared, such as a study where a plantar pressure measurement was conducted using 15 polymer fiber optic sensors in a 3D-printed insole in static and dynamic tests, obtaining that, with this development, it can also be used to measure body mass, in addition to being low cost and

having high portability [20]. Similarly, other authors created and evaluated a soft and flexible insole made of elastomeric materials with capacitive pressure measurement capabilities. This insole was designed to have four pressure-sensing zones in the contact area and was tested under different conditions showing reliable responses up to 300 kPa [21]. A similar approach was used in another study where electronic insoles were manufactured using 3D printing to measure plantar pressure. The authors used a microcontroller, and data was sent wirelessly to an app on an Android device in real-time. The force gradient across the insole was plotted in the app interface [22].

In this study, we examined if a digital manufacturing workflow for customized insoles can be implemented in middle-income countries, such as Peru, for peripheral neuropathy treatment, and if 3D-printed insoles perform as well as standard insoles. Here, we describe the development of customized insoles using 3D printing and semi-rigid filament materials as a proof of concept. These customized insoles were assessed using a clinical protocol and an in-shoe method based on a low-cost system with an open-source microcontroller and flexible sensors. The plantar pressure distribution and displacements were the main parameters for analysis.

## 2. Materials and Methods

For this study, scanning, modeling, and 3D printing were the selected digital manufacturing technologies in the fabrication workflow. Two filaments were used for the insole fabrication: thermoplastic polyether-polyurethane elastomer with additives (material A) and thermoplastic polyurethane polyester-based polymer (material B)—in both cases, the material is a polymer blend. These polymers are usually considered for medical device prototyping. For this proof of concept, a healthy adult male patient was recruited to participate in the fabrication workflow as a user and to evaluate insoles in walk trials. Plantar pressure distribution was analyzed quantitatively using an FSR-based system. Insoles effects were evaluated with a protocol adapted from Owings et al. [23] and Chambers and Sutherland [24]. The Institutional Bioethics Committee of the Universidad Peruana Cayetano Heredia approved this study (SIDISI: 102533). The patients' data were anonymized, stored in a database, and protected with a password to which only the researchers had access.

### A. Orthopedic insole development

The manufacturing workflow (Figure 1) is described in more detail:

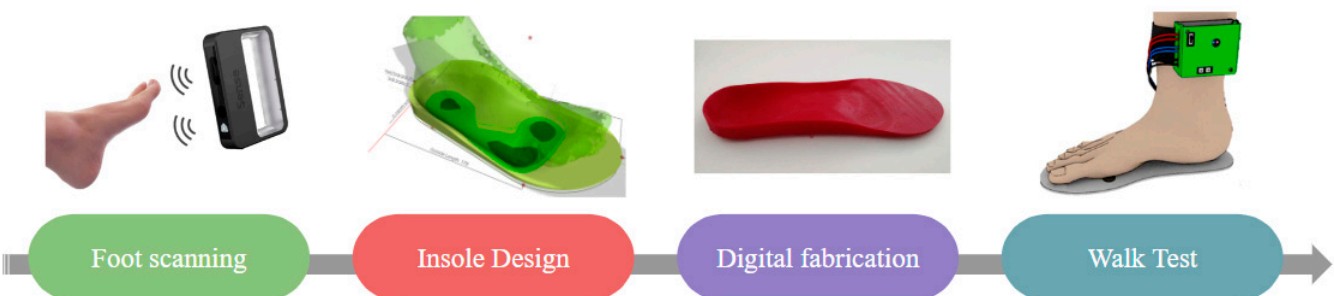

**Figure 1.** Digital manufacturing workflow for 3D-printed insoles.

#### A.1 Foot scan

The foot model was obtained at the first stage using a precision hand scanner, a Sense 2 model of 3D Systems. The scan was performed by placing the patient foot in a resting position and turning around. The scanning process took 4 min per foot. The distance between the scanner and the foot was 40 to 60 cm.

#### A.2. Insole design

Gensole software [25] was used for the second stage in insole modeling. This software is a web-based tool that allows for designing insoles that will be fabricated using 3D printing. The upper surface of the insole was shaped to match the foot model. An appropriate padding density was chosen, and contour curves were added to the insole to provide a

better fit within the shoes. For the creation of this insole, the following parameters were used within the program:

- Total insole length: 270 mm.
- Measurements at the ends of the insole: 25 and 10 mm.
- Infill up to 50% in the most extensive contact area, as shown in dark green in Figure 1.
- The minimum thickness of the insole was 3 mm.
- Previous steps were performed for the right and left foot.

More information about insole design can be seen in Appendix A.

### A.3. Digital Manufacturing

FDM technology and flexible filament were used for insole fabrication. The properties of these polymers are described in Table 1.

**Table 1.** Material comparison.

|  | **Material A** | **Material B** |
|---|---|---|
| **Color** | Green | Red |
| **Brand** | FilaFlex | NinjaFlex |
| **Type** | Thermoplastic polyether-polyurethane | Thermoplastic polyurethane polyester-based polymer |
| **Density (g/cm$^3$)** | 1.08 | 1.19 |
| **Hardness Shore A** | 70 | 85 |
| **Elongation at break (%)** | 900 | 660 |
| **Tensile strength (Mpa)** | 30 | 26 |

To 3D print insoles, AMF format files generated by the Gensole tool had to be converted to Gcode first. The Gcode is a set of instructions that indicates the coordinates and movements the 3D printer must perform to build a 3D object. For generating the Gcode, a laminating software called Repetier Host [26] was used. The printing parameters were defined—printing temperature at 225 °C, layer height at 0.2 mm, infill percentage at 40% with a hexagonal pattern, and no supports. Printer speed and retraction could vary according to the 3D printer. Gcode was created, exported, and this file was uploaded to the 3D printer using an SD card. An MD-6C 3D printer was used for insole fabrication. This 3D printer has a 300 × 200 × 500 mm printing size and uses up to 8 different thermoplastics. The bigger workspace allowed for fabricating the insoles without any technical issues. The two pairs of insoles were manufactured using both flexible materials.

### B. Plantar pressure reading system

A low-cost and portable system was used to read the plantar pressures. This electronic system was developed and tested by one of the authors in a previous project for studying diabetic neuropathy [27]. A development module called Arduino Nano, which contains a microcontroller called Atmega328p, was the core for data reading and processing. Other relevant components were the piezoelectric sensors and the Bluetooth module. Three sensors were placed on each foot, and their distribution can be seen in Figure 2B. According to the literature, common sites for ulcerations in diabetes patients are the plantar metatarsal heads and the heel [13,27,28]. Due to this incidence of ulceration, we decided to place the three sensors in those locations to measure the plantar pressure. A 3D-printed case was made for user safety. All electronics components and a 3.7-volt rechargeable battery were inside the case. This system was placed in the ankle of the test subject using Velcro straps for physical exercises. The lightweight design, wireless communication, and rechargeable capacity made the system suitable for walk trials. More information about the system can be seen in Appendices B and C.

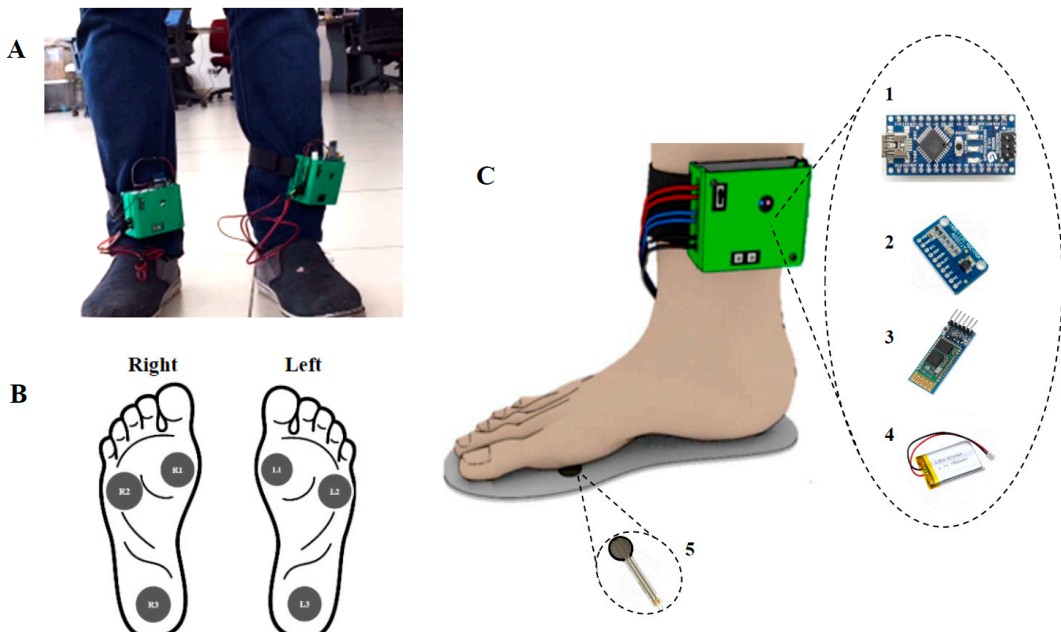

**Figure 2.** Plantar pressure evaluation. In (**A**), the test subject has an electronic system for the readings. In (**B**), the sensor distribution is shown on feet. In (**C**), the render represents how the system would be placed, and also shows its electronic components, such as (1) Arduino Nano: board with a microcontroller; (2) ADS1115: Board with digital analog converter channels; (3) HC-05: Bluetooth module; (4) LiPo 523450: Battery; and (5) FSR-402: Flexible pressure sensor.

For the measurement of the plantar pressure, an FSR-402 was used, which is a high-sensitivity resistive pressure sensor, for the configuration and calibration of this sensor, the equivalence table from the sensor datasheet was used [15]. In this technical document, resistance variations are shown in different graphs and allow users an appropriate configuration according to the application. The FSR-402 sensor was used in pull-down mode with a 10 kΩ resistance for the voltage divider. After this configuration, a new equivalence table can be generated, including pressure. Table 2 shows how resistance is translated to pressure units. For this table, the resistance of the sensor in units of kilo ohms (kΩ), the voltage of the voltage divider in volts (V), and the pressure measured in Newtons per square centimeter ($N/cm^2$).

**Table 2.** Equivalence between resistance, voltage, and pressure.

| FSR (Sensor Resistance) [kΩ] | FSR + R (Voltage Divider Resistance) [kΩ] | Current (mA) | Voltage (V) | Pressure ($N/cm^2$) |
|---|---|---|---|---|
| 30 | 40 | 0.095 | 0.95 | 0.2 |
| 6 | 16 | 0.238 | 2.37 | 1 |
| 1 | 11 | 0.347 | 3.47 | 10 |
| 0.25 | 1.25 | 0.372 | 3.72 | 100 |

## C. Experimental Setup

The 3D-printed insoles were fabricated using the dimensions of the patient's feet and the manufacturing workflow described above. We designed a simple clinical protocol that gives enough information for design purposes: foot and postural examination, observational gait analysis, and interview. Pressure distributions were recorded with the in-shoe electronic system in walk trials.

The type of foot (planus, cavus, normal) and foot deformities were observed and defined for foot examination. Hip, knee, and ankle joint mobility and alignment were assessed for postural examination. Observational gait analysis was performed indoors on a

concrete floor. The test subject used standard socks and shoes and performed three trials of single round-trip walks (180 s) using different insoles: flat standard insoles, insoles made with material A, and insoles made with material B. Patient walked back and forth in front of a physiotherapist to allow evaluation of hip, knee, and ankle. Videos were recorded for the three walk trials to facilitate analysis. References points were defined in video frames to identify axial or rotational abnormalities in the patient's walk. For this study, the patient chose his own walking speed during trials. This self-selected speed should represent conditions typically in a patient's life [23].

Insole performance and the effects on load distribution were analyzed with the electronic system during walk trials. A field researcher placed the electronic system on the patient's leg and adjusted it using Velcro straps (Figure 2A,C). Then, the patient started to walk two lengths of 10 m (turn included). Before each walk trial, the electronic system was placed and calibrated in an initial double-limb support state. The recorded data was sent to a laptop wirelessly during the walking tests for further analysis. In Figure 3, you can see the data obtained from the 3 sensors of the right and left insole during the walk test using the standard insoles.

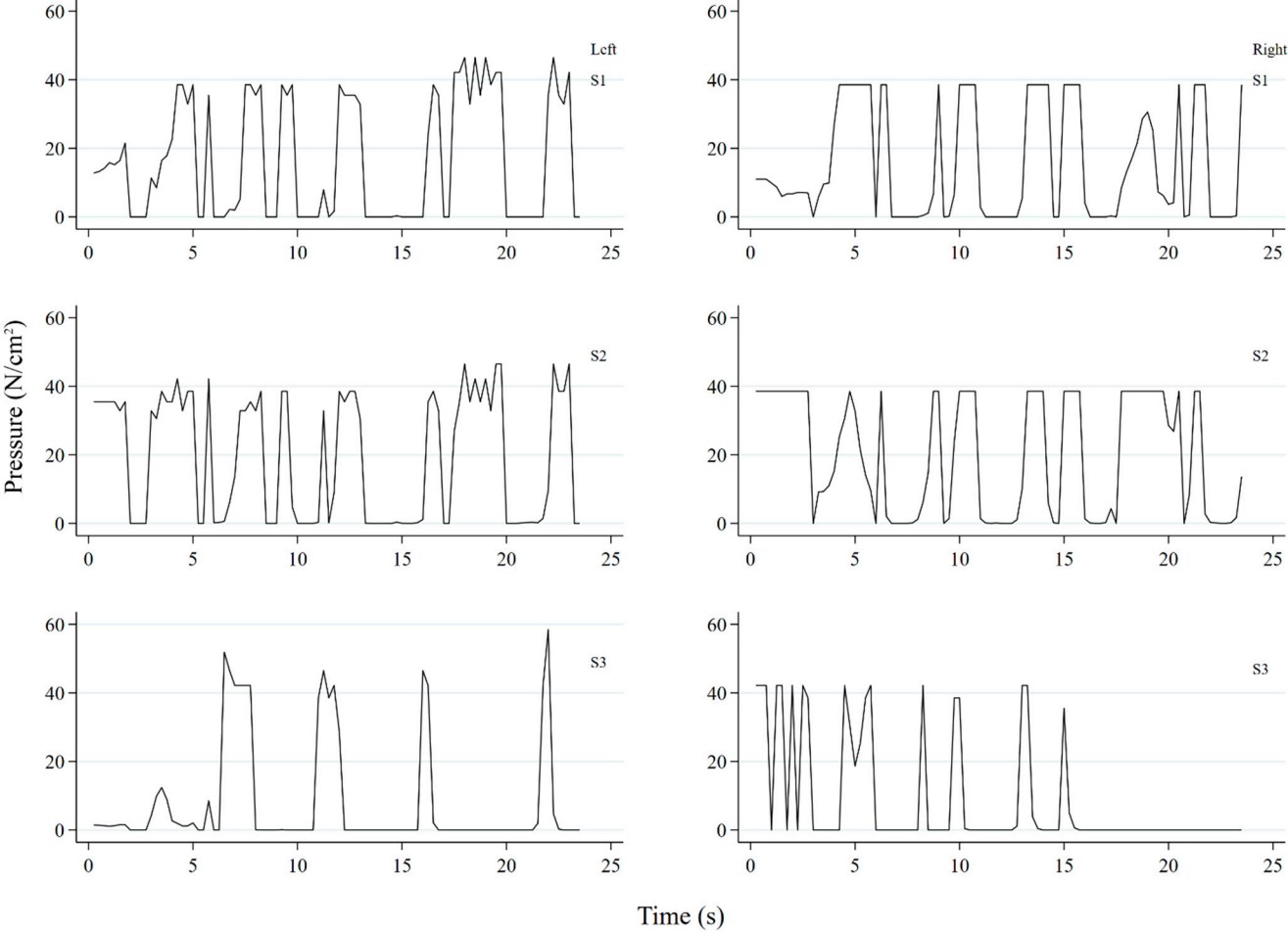

**Figure 3.** Values obtained from the FSR-based system in a walk trial with the standard insole.

Finally, a semi-structured interview was performed with the patient after each walk trial to gather user experience information. Questions were related to walking and comfortability.

## 3. Results

The manufacturing workflow began with foot scanning. The 3D model of this body part was appropriately reconstructed. Gensole generated the 3D model of the insole for

fabrication using FDM technology. The two pairs of insoles with flexible polymer were manufactured in less than two days. Of all the steps, 3D printing was the most time-consuming task in this workflow, spending almost 16 h for an insole pair. No redesign or post-processing of the insole was needed. The result of the insole made with material A can be seen in Figure 4. After insoles with material A and material B were manufactured, the patient used them and performed three walks, each with a different type of insole.

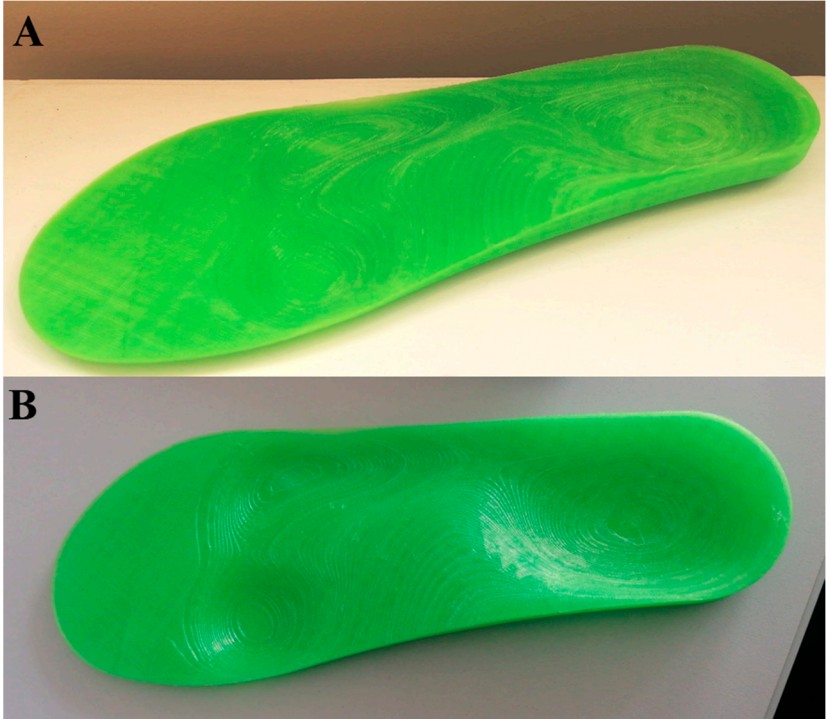

**Figure 4.** Right 3D-printed insole using thermoplastic polyether-polyurethane.

For the quantitative analysis, the average peak pressure was selected as the only parameter for evaluation, and one-way ANOVA tests were performed to compare the maximum pressure values obtained from each insole tested. The average from walk trials with standard insoles was the reference (45.11 N/cm$^2$). In the walk trial with material A insoles, the average was 47.49 N/cm$^2$ ($p = 0.188$). Meanwhile, the average with material B insoles was 42.56 N/cm$^2$ ($p = 0.085$). This information can be seen in Figure 5.

In observational gait analysis, variations in walk trials were identified at the stance phase. Based on the videos, material A insoles did not allow metatarsal heads to obtain appropriate contact with the concrete floor. Referent points on video frames to identify axial or rotational abnormalities can be seen in Figure 6. Stability was slightly modified and immediately compensated by the patient during the gait cycle. The step length was the minimum. In contrast, material B insoles allowed an effective walk similar to the standard insoles. The hip flexion, knee flexion, and ankle dorsiflexion were appropriated. Longer step length and faster walk speed were identified compared to the material A insoles. A video describing these results can be found as Supplementary Material. In interviews, the patient reported discomfort and pain in some steps with material A insoles. He felt its surface was more rigid than material B insoles and standard insoles. The pressure concentrations on metatarsal heads and heels in walk trials did not present an observable difference. No temporal data was plotted for this analysis.

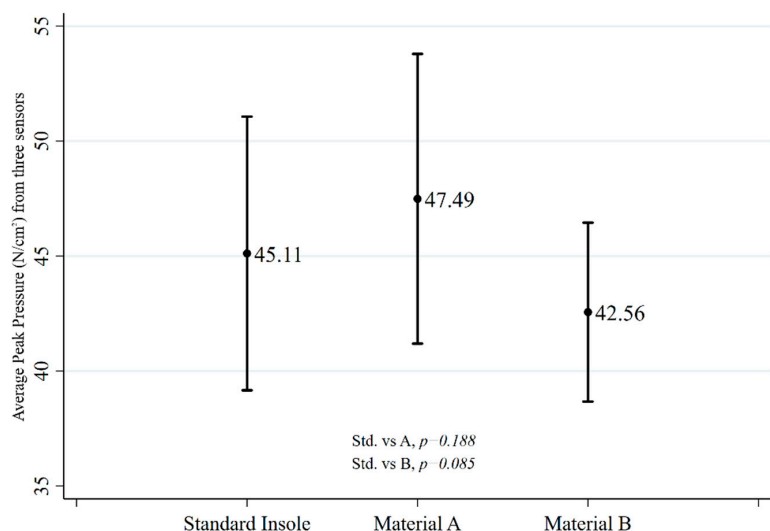

**Figure 5.** Average peak pressure for different insoles during the walk trials.

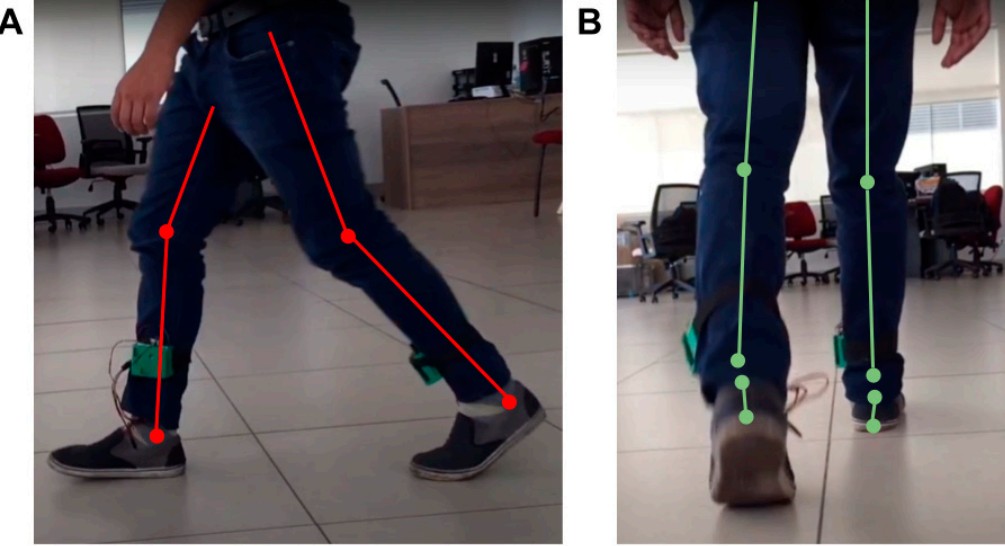

**Figure 6.** Gait analysis of the patient: (**A**) side view using standard insole and (**B**) back view using material A.

## 4. Discussion

Three-dimensional printing has become relevant for improving healthcare accessibility, especially for prevention and rehabilitation. Diabetes mellitus is a chronic disease with a higher prevalence in low- and middle-income countries. Patients with diabetes in these countries can benefit from 3D printing by obtaining customized insoles fabricated for the prevention or individual treatment of ulcers. However, most studies using this technology are still performed in developed countries, and the manufacturing workflow is difficult to reproduce, or information is under request. This study has demonstrated that 3D-printed insoles can be fabricated and validated in the Global South with an interdisciplinary approach. A manufacturing protocol has been well described for promoting reproducibility in the region.

In terms of the 3D-printed insoles, they performed as well as standard insoles, despite the mechanical properties of polymers. This outcome may be explained because no pressure redistribution modifications were performed during the insole design. Printing parameters could affect plantar pressure due to several configurations of internal patterns and density that can be established. Simulations and more points in the plantar surface

for measurements could be helpful for insole design. For in-shoe methods, several studies have sought to measure plantar pressure in flat insoles using different types of sensors, such as capacitive [21] and piezoelectric [22]. According to the systematic reviews [29], the latter is the most used. Capacitive sensors are usually built by research teams since they are not usually on the market. Their cost is relatively high due to the complexity of their components. As a result, reproducibility is limited, especially for low- and middle-income countries. Due to this context, we decide to develop an electronic system with piezoelectric sensors of the FSR family for plantar pressure measurements, making this system robust and accessible.

Finally, this study presents some limitations at the methodology level. A 3D commercial scanner was used at the scanning stage, which is still expensive for some low- and middle-income countries. Photogrammetry is a promising method for scanning body parts by taking pictures and creating 3D models. This technique has been used for fabricating facial prostheses, where scanning was performed using a smartphone [30]. We used photogrammetry for foot scanning, but 3D models were not as accurate as the commercial scanner. By creating and validating a scanning protocol and improving reconstruction techniques, a smartphone could replace commercial scanners making more accessible the insole fabrication workflow. At the digital fabrication stage, only two flexible filaments with the same printing parameters were used in the study. Mixed materials and different patterns and infill should be explored to understand their effects on plantar pressure distribution. Three-dimensionally printed filaments with novel antibacterial properties are interesting for ulceration prevention [31]. Validation of 3D-printed insoles was performed only with one participant.

Further studies should explore emerging technologies such as artificial intelligence. There are previous works where machine learning has been used in the evaluation of plantar pressure in insoles for the diagnosis of neuropathies in patients with diabetes [27], or where including these sensors has enabled researchers to use the data to analyze the gait and predict the ability to balance in real time [32]. These same techniques can be used to determine which regions of the insole can be modified in the 3D model to fabricate insoles that relieve pressure while the patient walks. More healthy participants should be included for clinical validation and to use insoles in real settings. Then, diabetic patients could be recruited for clinical trials or multicenter studies.

## 5. Conclusions

Diabetic neuropathy can lead to clinical complications such as ulceration or amputation of feet. For diabetes management, orthopedic insoles are needed to redistribute plantar pressure and prevent ulcer formation or allow ulcers to heal. Handmade insoles are usually used, but this traditional technique has limitations for customization. Three-dimensional printing is an emergent technology for insole fabrication, and several studies in developed countries have shown its potential benefits for individual treatment. We have demonstrated that a digital manufacturing workflow can be implemented for customized insole fabrication in middle-income countries such as Peru. A simple clinical protocol was proposed to gather user experience information.

Three-dimensionally printed insoles were manufactured using two different flexible polymers and fully adjusted to the foot compared to standard insoles, according to the patient report. Plantar pressures between foot and insoles, in the first and fifth metatarsal heads and heel, were measured in walk trials. Average peak pressures while using 3D-printed insoles presented no significant differences compared to standard insoles. In terms of user experience, patients reported that material B insoles were more comfortable than material A insoles and standard insoles. This preference was correlated with modifications in stability during walk trials.

**Supplementary Materials:** We have an explanatory video about observational gait analysis that can be downloaded at: https://www.mdpi.com/article/10.3390/designs6050095/s1, Video S1: Observational Gait Analysis.

**Author Contributions:** Conceptualization, M.L.-P., J.T.-M. and P.G.P.-H.; methodology, J.Z., M.M. and J.P.T.; software, J.Z. and J.P.T.; validation, W.P.-H. and M.M.; formal analysis, J.P.T.; investigation, M.L.-P., J.T.-M. and P.G.P.-H.; writing—original draft preparation, J.P.T.; writing—review and editing, J.Z., M.M., P.G.P.-H., M.L.-P., J.T.-M., W.P.-H. and J.P.T. All authors have read and agreed to the published version of the manuscript.

**Funding:** The main sponsors for this study were the Faculty of Medicine Alberto Hurtado (Fondo de Apoyo a la Investigación—2018) and the Department of Engineering at Universidad Peruana Cayetano Heredia.

**Institutional Review Board Statement:** The study was conducted in accordance with the Declaration of Helsinki, and approved by the Institutional Bioethics Committee of the Universidad Peruana Cayetano Heredia (SIDISI: 102533—Approved in November 2018).

**Informed Consent Statement:** Informed consent was obtained from all subjects involved in the study.

**Data Availability Statement:** The data that support the findings of this study are available from the corresponding author upon reasonable request.

**Acknowledgments:** Special thanks to the Department of Engineering at Universidad Peruana Cayetano Heredia for allowing access and use of equipment from its Digital Fabrication facility for manufacturing the 3D model prototypes.

**Conflicts of Interest:** The authors declare no conflict of interest.

## Appendix A. Insole Design

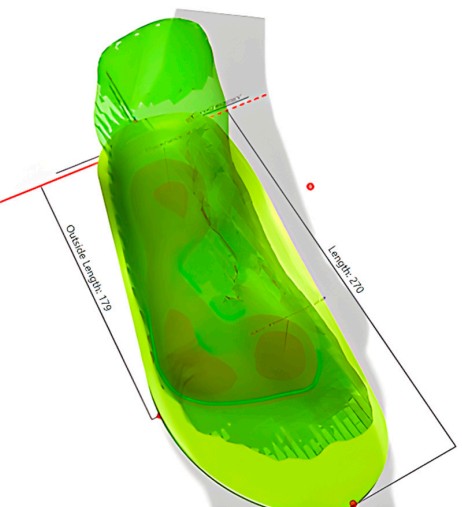

**Figure A1.** Total insole length.

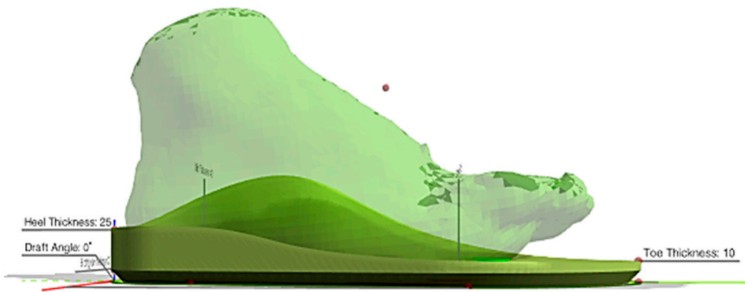

**Figure A2.** Measurements at the ends of the insole.

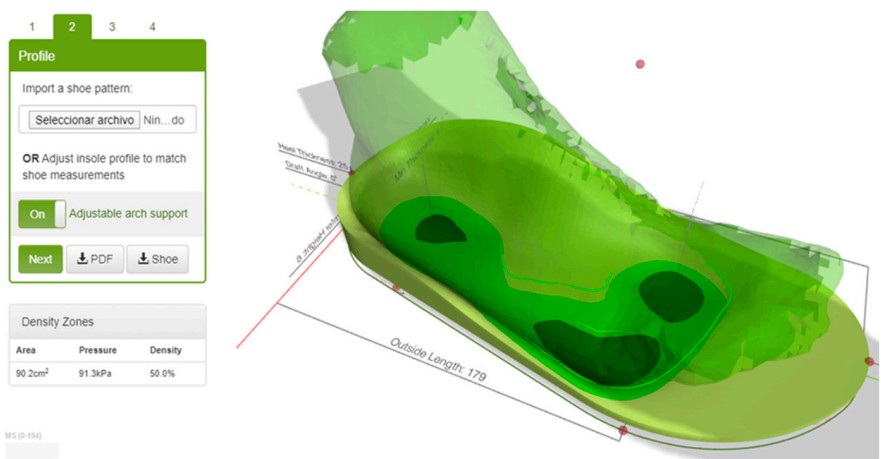

**Figure A3.** Insole infill.

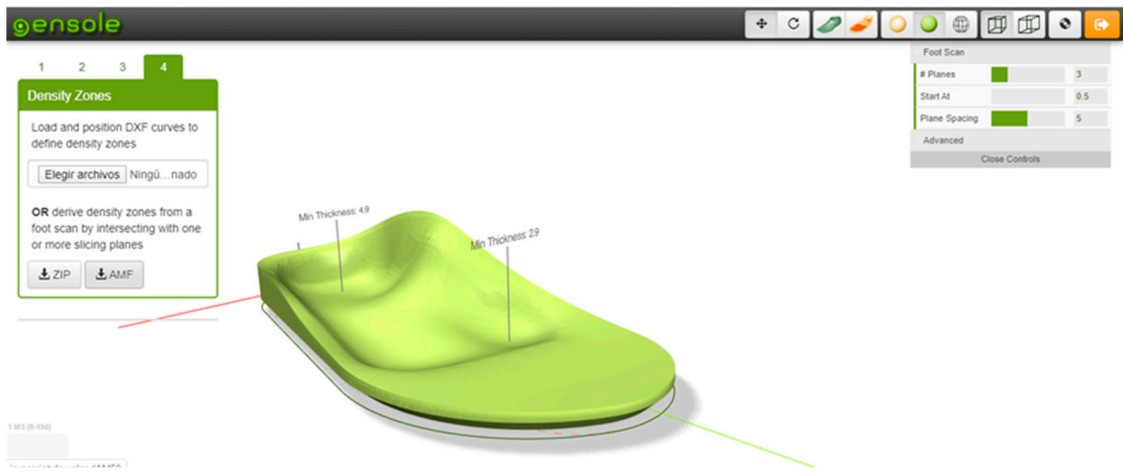

**Figure A4.** Minimum insole thickness.

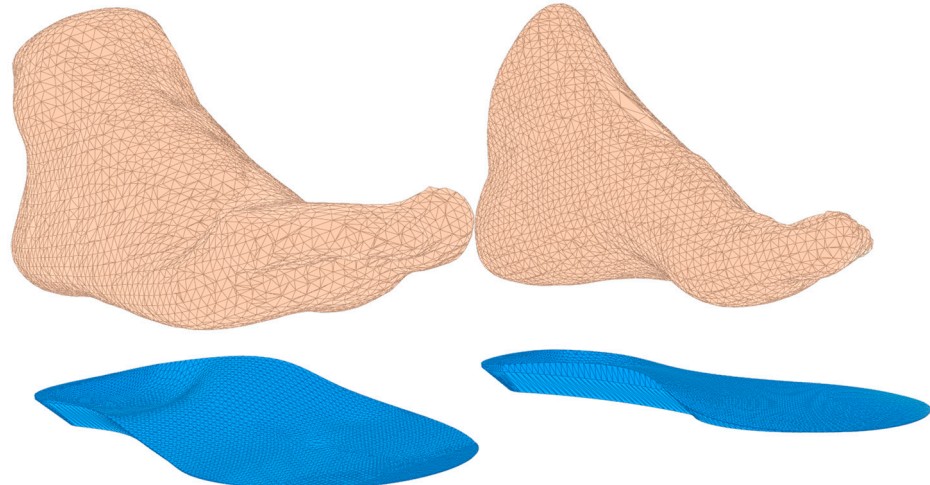

**Figure A5.** Scanned foot renders and insole design.

## Appendix B. Bill of Materials—Plantar Pressure Reading System

**Table A1.** Bill of materials.

| Item | QTY | Description |
|---|---|---|
| Arduino Nano | 1 | An 8-bit microcontroller has an eight-channel analog-digital converter (ADC) that allows it to simultaneously read different sensors and direct USB connection to programming and acquire data. |
| FSR-402 | 3 | A piezoelectric sensor presents a decrease in resistance as the force increases on its surface. |
| ADS1115 | 1 | ADC module with four channels of 16-bit resolution, the digital data is read through an I2C bus and can reach up to 860 samples per second. |
| HC-05 | 1 | These modules use a Bluetooth communications protocol specially designed for low-consumption devices. They are short-range since they have a maximum distance of 10 m. |
| Li-Po 523450 | 1 | Rechargeable lithium polymer battery with 1000 mAh and 3.7 volts. |

## Appendix C. Printed Circuit Board

A printed circuit board (PCB) was designed to use all the components in a real-time walk test. The ADS1115 module communicated with the microcontroller using I2C communication (pins A4 and A5 of the Arduino Nano). The values read by the sensors were sent using those pins.

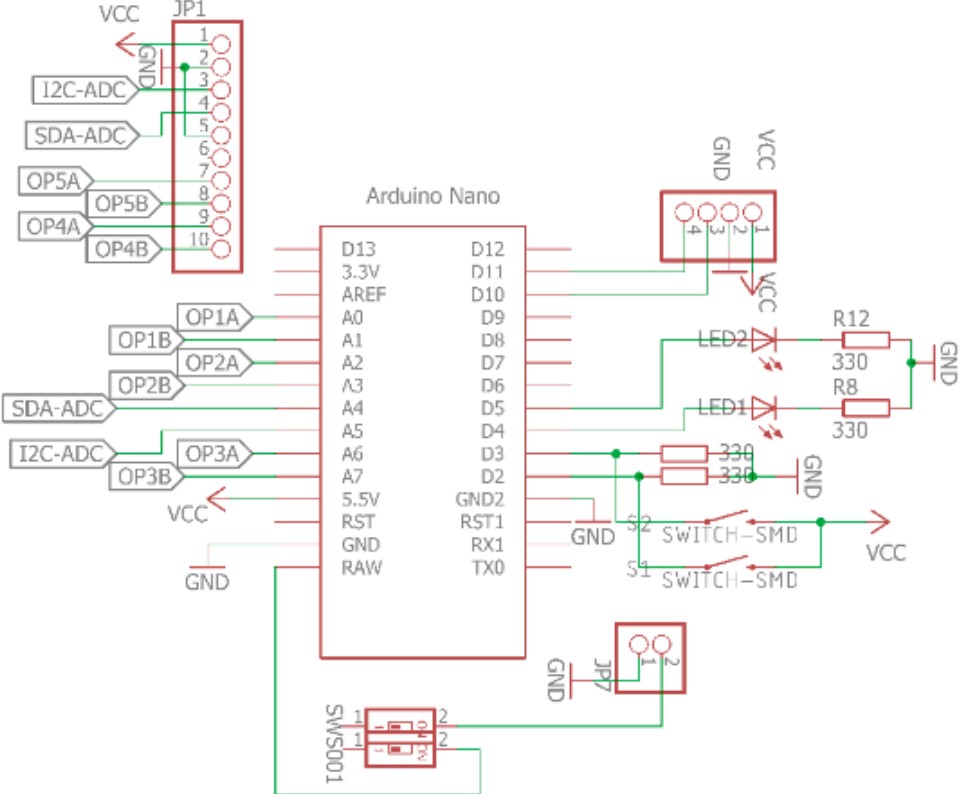

**Figure A6.** Circuit schematic design.

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
