# Peer review of "Development of 3D-Printed Orthopedic Insoles for Patients with Diabetes and Evaluation with Electronic Pressure Sensors"

_designs, 2022_

Round 1

Reviewer 1 Report

Journal: Designs by MDPI

Title: Development of 3D Printed Orthopedic Insoles for Patients with Diabetes and Evaluation with Electronic Pressure Sensors

Comments for Authors

        This study utilized 3D printing technology to manufacture customized insoles that better fit with feet. The pressure responses during testing walks were recorded by a wearable device. The motivation is interesting as, in human body, trivial imbalance may induce compensations elsewhere and result in distant physical malfunctions. However, this paper needs many other experiments and statistical analysis to support the benefit of using elastomeric customized insoles, and the discussion is quite limited, making the overall quality far from acceptance. As a result, a rejection is regretfully recommended. Please check the following comments.

1.     Thermoplastic elastomer is a more generic term including many categories like PU or polyisoprene. To better describe the details and compare the material properties with PU, “TPE” in this article should be replaced with the exact type of the material.

2.     What does “middle-resource settings” mean? Not clear. (Line 34)

3.     What does “on the effect of the same name” mean? I know this is talking about piezoelectric property, but the description here is not straightforward and clear (similar issue as Point 2 above). (Line 74)

4.     A typo of reference [120] which should be [20]. Using reference managing software is recommended. (Line 104)

5.     The maximum width of the insole is just 3 mm? (Line 144)

6.     Poor resolution of Figure A1-4. The words are blurred.

7.     How was the trip designed? Details such as the route, speed, and slope should be clearly mentioned for reproducing the experiment. And these parameters predominantly impact the pressure data.

8.     It is difficult to identify which sample corresponds to the figures. What do the legends mean? (Figure 3)

9.     The authors described the loading as pressures but demonstrated the data as forces. Please unify the terms, check the unit of force, and use SI units in all Figures. Not sure about what “g” here is because gram is the unit of mass and g-force is the unit of acceleration.

10.  The data in Figure 5 seem to have no significant difference. The averages here make less sense because of the widespread data. Statistical analysis should be categorized by maximum and minimum forces or any other definitions instead of averaging the overall data, and the averages and standard deviations in each category can be compared. The statistic in this paper is the major problem. There is also no statistical analysis in Figure 4. So far, the results don’t show robust evidence and the conclusion is questionable.

11.  What does the gait analysis mean? There is no further information other than Figure 6. Even Figure 6 itself is difficult to understand.

12.  There was only one participant in this study, which doesn’t provide comprehensive data of the forces. Biomechanics is a very complicated field, and walking patterns can vary a lot by person-to-person. Factors like ages, genders, and height/weight should also be considered and evaluated. Otherwise, with only one set of data, it is hard to conclude that the variance was sorely from the materials.

13.  There are numerous experiments should be included such as the validation to the sensor (as it is a non-standard, house-made model), materials characterizations like compressive modules (because you are using them as insoles adapted by pressures), and in-situ mapping of the pressure distribution. Other biomechanics experiments should also be involved.

14.  Many results are descriptive (Line 221-225). “The step length was the minimum” The plantar pressure. The flexions. The longer step length and faster walking speed. All of these results should be reported as numerical data, not just descriptions. The results are very limited to draw a conclusion now.

15.  The discussion is extremely short and only one other study has been cited and compared.

16.  Since this study involves testing on humans, please add IRB approval information if applicable.

Author Response

Response to Reviewer 1 Comments
Dear reviewer, here we send you the answers to the comments that you proposed. Additionally please review the new version of the manuscript attached to this response in a Word file.

Point 1:
Thermoplastic elastomer is a more generic term including many categories like PU or polyisoprene. To better describe the details and compare the material properties with PU, “TPE” in this article should be replaced with the exact type of the material.

Response 1: The term TPE was replaced by “Thermoplastic Polyether-Polyurethane elastomer with additives” in the article. Changes can be seen in line 30 in the final version of the manuscript. For mentioning the two polymers, material A and material B were used along the article.

Point 2: What does “middle-resource settings” mean? Not clear. (Line 34)

Response 2: “Middle-resource seetings” means settings in low- and middle income countries. We decided to update those words to make clear for readers. Changes can been seen in line 43 in the final version of the manuscript.

Point 3: What does “on the effect of the same name” mean? I know this is talking about piezoelectric property, but the description here is not straightforward and clear (similar issue as Point 2 above). (Line 74)

Response 3: The sentence was rewritten for better understanding in line 83 in the final version of the manuscript.

Point 4: A typo of reference [120] which should be [20]. Using reference managing software is recommended. (Line 104)

Response 4: This typo was a mistake made by the reference managing software and different versions of Microsoft Word. Yes, the reference should be 20.

Point 5: The maximum width of the insole is just 3 mm? (Line 144)

Response 5: The term was wrong. The correct term is the “minimum” value of thickness. Changes can be seen in line 159 in the final version of the manuscript.

Point 6: Poor resolution of Figure A1-4. The words are blurred.

Response 6: The Figure A1-4 was updated with a better resolution. Changes can be seen in line 360 in the final version of the manuscript.

Point 7: How was the trip designed? Details such as the route, speed, and slope should be clearly mentioned for reproducing the experiment. And these parameters predominantly impact the pressure data.

Response 7: The evaluation was performed based on a protocol adapted from Owings et al. and Chambers and Sutherland [Ref 23,24]. For this study, the patient selected their own walking speed during the tests. This self-selected speed should represent typical conditions in a patient's life [Ref 23]. Each walk trial took 180 seconds. The observational gait analysis was performed indoors on a concrete floor. “Materials and Methods" and "Experimental setup" sections were updated.

Point 8: It is difficult to identify which sample corresponds to the figures. What do the legends mean? (Figure 3)

Response 8: The figure was improved and updated, indicating which sensors are equivalent to the right or left side, it was also clarified in the description that this reading corresponds to the test with the standard insole. Changes can be seen in line 241 in the final version of the manuscript.

Point 9: The authors described the loading as pressures but demonstrated the data as forces. Please unify the terms, check the unit of force, and use SI units in all Figures. Not sure about what “g” here is because gram is the unit of mass and g-force is the unit of acceleration.

Response 9: A table has been added detailing the levels between the resistances and voltages obtained from the sensors with their equivalences in Newton units to express the pressure obtained. This table is on line 209 in the final version of the manuscript

Point 10: The data in Figure 5 seem to have no significant difference. The averages here make less sense because of the widespread data. Statistical analysis should be categorized by maximum and minimum forces or any other definitions instead of averaging the overall data, and the averages and standard deviations in each category can be compared. The statistic in this paper is the major problem. There is also no statistical analysis in Figure 4. So far, the results don’t show robust evidence and the conclusion is questionable.

Response 10: We have now performed an analysis considering the peak pressure detected by the sensors (the maximum value). It has allowed us to compare the average pressure obtained by the three sensors among each type of insole. Additionally, Figure 4 was eliminated and the conclusions were updated based on the new information. Despite the slight variations, there is no significant difference between 3D printed insoles and the standard insoles.

Point 11: What does the gait analysis mean? There is no further information other than Figure 6. Even Figure 6 itself is difficult to understand.

Response 11: We can describe "gait analysis" as the process of recording and interpreting biomechanical variables obtained while walking, which allows any alteration to be detected and supports the clinical decision-making process. In this case it is an observational gait analysis in which the patient walks back and forth in front of a physical therapist to allow evaluation of the hip, knee and ankle. Videos of the three walking tests were recorded to facilitate analysis. Landmarks were defined on video frames to identify axial or rotational abnormalities in the patient's gait. "Experimental setup" was updated.

Point 12: There was only one participant in this study, which doesn’t provide comprehensive data of the forces. Biomechanics is a very complicated field, and walking patterns can vary a lot by person-to-person. Factors like ages, genders, and height/weight should also be considered and evaluated. Otherwise, with only one set of data, it is hard to conclude that the variance was sorely from the materials.

Response 12: Due to the customized insoles developed in this study were a proof of concept, we decided to use qualitative methods (user experience and observational analysis) where only one participant was enough for prototyping and testing. The aim of this article is to develop and implement a fabrication workflow for customized insoles and biomechanics will be evaluated in further studies with a bigger sample size (Lines 21 to 240 in the final version of the manuscript)

Point 13: There are numerous experiments should be included such as the validation to the sensor (as it is a non-standard, house-made model), materials characterizations like compressive modules (because you are using them as insoles adapted by pressures), and in-situ mapping of the pressure distribution. Other biomechanics experiments should also be involved.

Response 13: This study is a proof of concept that proposes tools and a workflow for insole design and fabrication in low- and middle-income countries. We considered appropriate using an observational gait analysis and in-shoe methods using low-cost electronic systems. For biomechanics analysis, highly specialized laboratories are still needed and limit reproducibility. They should be included in further studies to include more variables in the evaluations. The pressure sensors were validated in a previous publication (Ref 22).

 Point 14: Many results are descriptive (Line 221-225). “The step length was the minimum” The plantar pressure. The flexions. The longer step length and faster walking speed. All of these results should be reported as numerical data, not just descriptions. The results are very limited to draw a conclusion now.

Response 14: In this work, the observational analysis of the gait was chosen, this was also complemented by an interview where the patient could give qualitative descriptions of his experience as a user, for this reason this section of the results is descriptive. “Results” section was updated.

Point 15: The discussion is extremely short and only one other study has been cited and compared.

Response 15: The "Discussion" section was rewritten. The new version discusses new points of view on the development of the project such as the manufacturing process, the pressure measurement method, the context of research in our region, future work and limitations, citing 7 articles.

 Point 16: Since this study involves testing on humans, please add IRB approval information if applicable.

Response 16: Yes, we have permission from the ethics committee of the Peruvian university Cayetano Heredia and this was added on line 138 in the final version of the manuscript.

For all this we hope you can reconsider the initial decision you had to reject this publication, since we believe that we have been able to improve in all the aspects that you requested, we hope that this new version is sufficiently improved for its subsequent publication

Reviewer 2 Report

This manuscript reported sensor for diabete and pressure sensing. The topic is ineresting and useful. I recomment to publish before the following issues are solved.

1, To solid 3D printing by using soft materials, the authors can cite more papers to solid the background, such as JMM 32(2022) 064004

2, As to the design by 3D printing, why the reason to choose the materials?

3, What is the reason for pressure sensing in Figure 2? any theory?

4, What is the sampling rate in Figure 3? did the author design the force sensor?

5, The walking test is not enough, the author can list more test data in this manuscript to solid the sensing system

Author Response

Response to Reviewer 2 Comments
Dear reviewer, here we send you the answers to the comments that you proposed. Additionally please review the new version of the manuscript attached to this response in a Word file.

Point 1: To solid 3D printing by using soft materials, the authors can cite more papers to solid the background, such as JMM 32(2022) 064004

Response 1: We consider that the paper suggested its not related to fused deposition modeling  technology pointed out in the article. Other articles have been added: (Ref 29 to 32)

Point 2: As to the design by 3D printing, why the reason to choose the materials?

Response 2: There are antecedents referenced in the paper where the use of these semi-rigid filaments is described, (Ref 13), in addition to the fact that these materials are the most commercially available in the local market of our country.

Point 3: What is the reason for pressure sensing in Figure 2? any theory?

Response 3:

  • For plantar pressure measurements, an in-shoe method based on sensors was created in the sixties (Ref 13). Over the years, several in-shoes methods have been developed (Ref 14) and researchers commonly use them with diabetes patients to study ulceration and plantar pressures in lab and real settings (Ref 23).
  • According to the literature, the common sites for ulcerations in diabetes patients are the plantar metatarsal heads and the heel. Due to this incidence of ulceration, we decided to place three sensors in those locations to measure the pressure (Ref 14 and 18).

Point 4: What is the sampling rate in Figure 3? did the author design the force sensor?

Response 4: The data reading was done at 4 samples per second, the sensor is a commercial version (FSR-402), the data sheet can be found in reference 15, only the PCB where all the components are connected was designed, it also describes the measurement curve and its equivalence in pressure, this section has been expanded in this new version of the article between lines 209 to 210 in the final version of the manuscript.

Point 5: The walking test is not enough, the author can list more test data in this manuscript to solid the sensing system

Response 5: There are multiple antecedents in the literature where, for example, they make a characterization of the sensor (ref 22) which has been taken as a reference to develop the equivalence table based on our design and there are also antecedents where they use the sensor to evaluate the plantar pressure of similarly in a standard template (ref 21)that also serve as a guide to test the templates created for this work

Without more to add, we hope that this new version is sufficiently improved for its subsequent publication.

Round 2

Reviewer 1 Report

Journal: Designs by MDPI

Title: Development of 3D Printed Orthopedic Insoles for Patients with Diabetes and Evaluation with Electronic Pressure Sensors

Comments for Authors

        The authors provided a revised version addressing the previous comments and clarifying the motivation and objective of this study. Although many other biomechanical experiments are needed to deliver conclusive results; however, as a proof of concept at the early stage, the manuscript can be proceeded to the next editing step after some suggested minor revisions below.

1.     Is polyether-polyurethane (Material A) a polymer blend, copolymer, or polyurethane with PEO-based soft segment? Need to specify. And also mention the type of soft segment of Material B if Material A is poly(ether)urethane.

2.     The unit of pressure should be Newton per unit area. N, Newton, is the unit of force. Please check all units.

3.     The abbreviation of kilo is lower case “k” (kW). Capital K is Kelvin temperature. Please check all applicable typos.

4.     Suggest specifying the material of the customized insole in Figure 6 and also uploading gait analysis video as the supplemental information to better demonstrate how the analysis worked. Trimmed videos are also fine to highlight the parts analyzed.

Author Response

Response to Reviewer 1 Comments

Dear reviewer, here we send you the answers to the comments that you proposed and we attach the version in track changes.

Point 1: Is polyether-polyurethane (Material A) a polymer blend, copolymer, or polyurethane with PEO-based soft segment? Need to specify. And also mention the type of soft segment of Material B if Material A is poly(ether)urethane.

Response 1: We have made the specification that the type of material is a polymer blend and according to his datasheet material B is a Thermoplastic Polyurethane based on polyester.

Point 2: The unit of pressure should be Newton per unit area. N, Newton, is the unit of force. Please check all units

Response 2: The unit of pressure has been modified by Newton per square centimeter (N/cm2). This change was made in all the parts where it was mentioned as "materials and methods", "results" and in the figures.

Point 3: The abbreviation of kilo is lower case “k” (k ). Capital K is Kelvin temperature. Please check all applicable typos.

Response 3: The use of the "K" was corrected for a lowercase one in all the places that were mentioned, in the same way "Kohm" was changed for "kΩ"

Point 4: Suggest specifying the material of the customized insole in Figure 6 and also uploading gait analysis video as the supplemental information to better demonstrate how the analysis worked. Trimmed videos are also fine to highlight the parts analyzed.

Response 4: It has been indicated to which insole each frame of the figure corresponds, we have also added an explanatory video on gait analysis as supplementary material

Without more to add, we hope that this new version is sufficiently improved for its subsequent publication.

Reviewer 2 Report

This manuscript solved all my issues, I recommet to publish in the current form

Author Response

We appreciate that the reviewer indicates that all issues have been resolved, we hope the prompt publication of the article
